# Dimethyl Fumarate-Loaded Gellan Gum Hydrogels Can Reduce In Vitro Chemokine Expression in Oral Cells

**DOI:** 10.3390/ijms25179485

**Published:** 2024-08-31

**Authors:** Lei Wang, Natalia dos Santos Sanches, Layla Panahipour, Atefe Imani, Yili Yao, Yan Zhang, Lingli Li, Reinhard Gruber

**Affiliations:** 1Department of Oral Biology, University Clinic of Dentistry, Medical University of Vienna, 1090 Vienna, Austria; wanglei20111213@163.com (L.W.); natalia.s.sanches@unesp.br (N.d.S.S.); layla.panahipour@meduniwien.ac.at (L.P.); dr_a_imani@hotmail.com (A.I.); 2Wenzhou Institute, University of Chinese Academy of Sciences, Wenzhou 325011, China; yanzhang@ucas.ac.cn (Y.Z.); lingli_lee@yeah.net (L.L.); 3Department of Diagnosis and Surgery, Araçatuba Dental School of Sao Paulo, Sao Paulo 16015-050, Brazil; 4State Key Laboratory of Ophthalmology, Optometry and Visual Science, Wenzhou Medical University, Wenzhou 325027, China; yylys917@163.com; 5Department of Periodontology, School of Dental Medicine, University of Bern, 3010 Bern, Switzerland; 6Austrian Cluster for Tissue Regeneration, 1200 Vienna, Austria

**Keywords:** dimethyl fumarate, gellan gum hydrogel, chemokine expression, oral cells, ERK, JNK inflammatory response

## Abstract

Dimethyl fumarate (DMF), originally proposed to treat multiple sclerosis, is considered to have a spectrum of anti-inflammatory effects that effectively control periodontitis, mainly when applied with a hydrogel delivery system. Chemokine expression by gingival fibroblasts is a significant driver of periodontitis; thus, hydrogel-based strategies to deliver DMF, which in turn dampen chemokine expression, are of potential clinical relevance. To test this approach, we have established a bioassay where chemokine expression is induced by exposing gingival fibroblast to IL1β and TNFα, or with saliva. We show herein that DMF effectively reduced the expression of CXCL8, CXCL1, CXCL2, and CCL2—and lowered the phosphorylation of ERK and JNK—without affecting cell viability. This observation was confirmed by immunoassays with CXCL8. Consistently, the forced chemokine expression in HSC2 oral squamous epithelial cells was greatly diminished by DMF. To implement our hydrogel-based delivery system, gingival fibroblasts were cocultured with gellan gum hydrogels enriched for DMF. In support of our strategy, DMF-enriched gellan gum hydrogels significantly reduced the forced chemokine expression in gingival fibroblasts. Our data suggest that DMF exerts its anti-inflammatory activity in periodontal cells when released from gellan gum hydrogels, suggesting a potential clinical relevance to control overshooting chemokine expression under chronic inflammatory conditions.

## 1. Introduction

Periodontal disease is caused by a chronic inflammation that progresses towards tissue destruction, with its clinical hallmarks including bleeding upon probing and other cardinal signs of inflammation [1]. Chronic inflammation not only destroys the periodontal soft tissue attachment but also culminates in the resorption of alveolar bone [2]. If untreated, the periodontal pockets expand and, ultimately, tooth loss is the consequence [3]. Periodontitis and peri-implantitis originate from a less severe form of mucositis, the latter being reversible when oral hygiene is considered [4]. Thus, the overall therapeutic goal is to remove, or at least control, the leading causes of inflammation, namely the dental plaque, which consists of a complex biofilm [5]. The logical first step in periodontitis and mucositis therapy is scaling and root planing to remove the biofilm [6,7]—and, ideally, to dampen the local inflammatory process that has recently been characterized on a cellular and molecular level by single cell RNA sequencing [8]. Here, gingival fibroblasts have been recognized as a major driver of chemokine-mediated neutrophil influx in periodontitis [8]. A similar cellular mechanism is linked to the inflammatory osteolysis of peri-implantitis [9]. Drug delivery strategies targeting the overshooting chemokine expression by fibroblasts are therefore a feasible therapeutic approach.

Dimethyl fumarate (DMF) is the methyl ester of fumaric acid and is named after Fumaria officinalis. It has pleiotropic effects on the cellular level; for instance, DMF exhibits anti-inflammatory effect and reduces oxidative stress [10]. Based on these and perhaps other beneficial properties, DMF was agreed by the Food and Drug Administration (FDA) as a treatment option for adults with relapsing multiple sclerosis and psoriasis under the trade name Tecfidera^®^ [11] and Skilarence^®^ [12], respectively. However, the spectrum of possible clinical applications of DMF extends towards different pathologies; for instance, cardiovascular, neurodegenerative, ocular, and gastrointestinal diseases, as well as tumors [13]. Inspired by the beneficial clinical effects that can be linked to the anti-inflammatory activity of DMF, we propose here the application of DMF in the context of a periodontitis and peri-implantitis therapy. The first step towards this clinical strategy is to understand the impact of DMF on modulating the inflammatory response of oral cells in vitro.

In vitro bioassays support the potential of DMF to reduce an inflammatory response. For instance, DMF inhibited agonist-induced cytokine secretion or other inflammatory reactions by fibroblasts [14], smooth muscle cells [15], endothelial cells [16,17], and epithelial cells [18]. DMF also reduced the forced matrix metalloproteinase expression in chondrocytes [19]. This anti-inflammatory activity is accompanied by an inhibition of MAPK signaling; for instance, DMF inhibits ERK and JNK phosphorylation, but not p38, in TLR-signaling-induced murine macrophages [20], and ERK phosphorylation in LPS-activated microglia cells [21]. Surprisingly, however, only limited studies are available confirming the anti-inflammatory response of DMF in oral cells. For instance, DMF lowered the LPS-induced pyroptosis in gingival and pulp fibroblasts [22]. Moreover, in oral squamous carcinoma cell lines, DMF modulates apoptosis, oxidative stress, and epithelial-mesenchymal transformation [23]. Bioassays related to the expression of chemokines are not available. There is obviously a demand to evaluate how DMF potentially reduces chemokine expression in oral cells. Considering that chemokine expression, including CXCL1, CXCL2, and CXCL8, heavily increased in periodontal disease [8], as well as peri-implantitis [9], we propose DMF can be applied to periodontal and peri-implant defects. For this aim, we propose an injectable, in situ-forming drug delivery system with hydrogel properties.

Gellan gum (GG) is a natural anionic FDA-approved polysaccharide polymer used to prepare in situ-forming hydrogels that serve as a potential drug delivery system, as summarized in a series of excellent reviews [24,25,26,27]. Hydrogels prepared from gellan gum have good biocompatibility, low degradability, and controlled mechanical properties. Easy access, low production cost, and non-toxic degradation products accelerate gellan gum usage in pharmaceutical industries [28]. We take advantage of gellan gum, a thermally sensitive hydrogel, being liquid and injectable when kept at room temperature [29]. Thus, when liquid gellan gum reaches body temperature, it becomes a hydrogel. It is therefore not surprising that gellan gum has attracted considerable attention in biomedical research, especially in drug delivery, tissue engineering, and wound healing [24,25,26,27]. For instance, gellan gum can deliver caffeic acid phenethyl ester, showing effective activity against C. albicans, making it suitable for use in stomatitis [30]. Moreover, gellan gum was proposed as a delivery system for dexamethasone [31], piroxicam [32], naproxen [33], and diclofenac [34]. Nevertheless, the potential of gellan gum serving as a delivery system for DMF has yet to be proposed.

The overall aim of the present study is thus twofold: first to evaluate DMF as a potential strategy to lower the forced-expression of chemokines in gingival fibroblasts; and second, to support the use of gellan gum as a DMF delivery system, with potential local applications in periodontal and peri-implant defects.

## 2. Results

### 2.1. Screening of DMF Concentrations

To understand the impact of DMF on cell viability, we analyzed the ability of gingival fibroblasts to convert a tetrazole substrate into formazan. As shown in Figure 1, the viability of gingival fibroblasts was maintained up to 100 μM DMF; however, there was a sharp drop at 200 μM DMF. HSC2 cells were more robust, showing regular formazan formation at 200 μM DMF. Green staining in the live/dead analysis further confirms that cells remain vital at 100 μM DMF (Figure 1). Ethanol (70%) was used as a positive control, showing red dead cells. Trypan blue (Appendix A) showed consistency with live/dead staining.

### 2.2. DMF Reduced IL1β and TNFα or SLV-Induced Chemokines in Gingival Fibroblasts

To provoke an inflammatory response, gingival fibroblasts were exposed to IL1β and TNFα, and saliva (SLV). As expected, IL1β and TNFα [35] and SLV [36] caused a robust increase in CXCL1, CXCL2, CXCL8, and CCL2 expression in the gingival fibroblasts. When gingival fibroblasts were simultaneously exposed to 100 μM DMF, the forced expression of CXCL1, CXCL2, CXCL8, and CCL2 significantly reduced (Figure 2). DMF at 50 μM had no significant effect (Appendix A). Gene transcription was supported by an immunoassay showing reduced CXCL8 on the translational level (Figure 2). Consistently, 100 μM DMF reduced the SLV-induced expression of CXCL1, CXCL2, CXCL8, and CCL2 (Figure 3). This observation was supported by an immunoassay for CXCL8 (Figure 3). However, DMF did not affect the NFκB-p65 nuclear translocation (Appendix A). Together, these findings suggest that IL1β and TNFα- and SLV-exposed gingival fibroblasts are a valid bioassay for chemokine expression and that DMF can reduce our selected panel of chemokines.

### 2.3. DMF Reduced IL1β and TNFα-Induced Chemokines in HSC2 Cells

We next used our established bioassay where IL1β and TNFα can induce chemokine expression in HSC2 cells [37]. A robust increase in CXCL1, CXCL2, CXCL8, and CXCL10 expression was detected that was significantly reduced by 100 μM DMF (Figure 4). These observations are supported by an immunoassay for CXCL8 (Figure 4). However, 50 μM of DMF failed to have a significant effect on chemokine expression (Appendix A).

### 2.4. DMF Inhibits the Phosphorylation of ERK and JNK, but Not of p38 or p65, in IL1β and TNFα-Induced Gingival Fibroblasts

To clarify the molecular mechanism underlying the anti-inflammatory effects of DMF, we analyzed the phosphorylation of ERK, JNK, p38, and p65 using Western blotting. As shown in Figure 5, IL1β and TNFα remarkably elevated the phosphorylation of ERK, JNK, p38, and p65. Notably, the phosphorylation of ERK, and to some extend also JNK, was attenuated by 100 μM DMF, where the phosphorylation of p38 and p65 were not affected.

### 2.5. Rheological Properties of Gellan Gum Hydrogel Loaded with DMF

To prepare the hydrogels, gellan gum powder was dissolved in distilled water to reach concentrations ranging from 0.5% to 1.25%. The homogeneous and translucent liquid gellan gum was measured using a rheometer. As shown in Figure 6, 0.75% gellan gum failed to form a hydrogel as, under a progressively increasing temperature, the storage modulus (G′) was lower than loss modulus (G″). However, 1% gellan gum showed a hydrogel-like behavior when the temperature increased to 30 °C, as G′ became larger than G″. The 1.25% gellan gum already gelled at 25 °C, which is not ideal for injection. Therefore, we used 1% gellan gum to prepare the injectable self-curing hydrogel. Loading the hydrogel to reach a final concentration of 0.5 mM (LG) or 1.0 mM (HG) of DMF maintained the rheological properties, where the G′ was higher than G″ at 37 °C. Thus, we considered the 1% gellan gum hydrogel to be a suitable DMF delivery system to be implemented in our chemokine expression bioassay.

### 2.6. Morphology, Injectability and Swelling Properties of Gellan Gum Hydrogel Loaded with DMF

Next, we evaluated further physicochemical properties of the 1% gellan gum hydrogel (GG) loaded with 0.5 mM (LG) or 1.0 mM (HG) DMF. SEM images of the hydrogels showed relatively homogeneous microporous and interconnected structures (Figure 7A). The average pore size, however, is lowered by the presence of DMF, with 54.5 ± 14.56 μm in GG compared to 20.16 ± 8.27 μm and 26.41 ± 5.35 μm in LG and HG, respectively. Injectability was confirmed by squeezing the 1% gellan gum liquid through a 26G needle (Figure 7B). The gellan gum liquid could be injected and fit well to the spaces between the magnetic bead, showing injectability of the hydrogel. To evaluate the swelling behavior of the hydrogels, their weight increase at different time points was measured. As shown in Figure 7C, after being immersed in PBS for 24 h, the swelling ratio was between 20% to 40%, and reached an equilibrium state at 72 h, which did not vary significantly among LG, HG, and GG. After 72 h, the weight began to decrease and the hydrogel started to degrade.

### 2.7. DMF Release

The potential of the gellan gum hydrogel in tuning the DMF release was observed in PBS at 37 °C (Figure 8). DMF was burst-released from the gellan gum hydrogel in both LG and HG groups during the first 8 h, with release rate of 90.8 ± 6.7% and 91.2 ± 9.6%, respectively. The burst release of DMF from the gellan gum hydrogels could be explained by the Ficken diffusion mechanism [38,39]. The DMF release slowed down over the next 5 days. The cumulative release of LG and HG were 64.6 ± 0.8% and 58.5 ± 3.2%, respectively, at 120 h. The cumulative release decreased after 8 h, which could be due to the rapid hydrolysis of DMF to monomethyl fumarate (MMF) in phosphate buffer, pH 7.4 [40,41].

### 2.8. DMF Released from Hydrogel Reduced IL1β and TNFα-Induced Chemokines in Gingival Fibroblasts

To further analyze the DMF release from the gellan gum hydrogel, we performed a coculture bioassay with the gingival fibroblasts (Figure 9). The live/dead staining also verified the safety of using DMF-loaded gellan gum hydrogel in the cell model. We used IL1β and TNFα to induce the inflammatory response of gingival fibroblasts, followed by stimulation of the DMF-loaded hydrogel. The RT-PCR results showed a robust increase in the CXCL1, CXCL2, CXCL8, and CCL2 expression when the cells were exposed to IL1β and TNFα. When the gingival fibroblasts were simultaneously exposed to IL1β and TNFα, and a DMF-loaded hydrogel, the forced expression of CXCL1, CXCL2, CXCL8, and CCL2 significantly reduced (Figure 10). These findings were supported by an ELISA showing CXCL8 reduction by the DMF-loaded hydrogel (Figure 10). These findings suggest that DMF was released from the hydrogel scaffold and remained active.

## 3. Discussion

Inflammation is related to many chronic diseases, such as periodontitis [1]. Tremendous research efforts have been undertaken to uncover the molecular mechanisms of inflammatory responses in periodontal diseases [8]. Gingival fibroblasts and oral epithelial cells were identified to be significant drivers of inflammation in periodontitis [8] and periimplantitis [9]. Nevertheless, how these cells respond to modulators of inflammation has yet to be fully understood. DMF has inflammation modulatory effects in various in vitro bioassays [14,15,16,17,18,19], but research on the impact on gingival fibroblasts and oral epithelial cells has only just started [22,23]. In the present study, IL1β and TNFα caused a robust increase of chemokines in gingival fibroblasts and the oral squamous carcinoma cell line HSC2 cells, consistent with our previous findings [35,37]. In addition, saliva increased chemokines in gingival fibroblasts, including CXCL1, CXCL2, CXCL8, and CCL2, which also supported our observations [36]. However, one significant finding of the present research was that DMF exerted an inhibitory effect on the expression of inflammatory mediators in gingival fibroblasts and HSC2. Treating gingival fibroblasts and HSC2 cells with DMF reduced the inflammatory expression of chemokines, a mechanism that can be linked to the decrease of ERK and somewhat also JNK signaling; consistent with what has been observed with murine macrophages [20] and microglia cell [21], but surprisingly not obviously affecting NFκB-p65 signaling [20]. Thus, these investigations provide a fundamental insight into how DMF modulates inflammatory signaling pathways in an in vitro setting with gingival fibroblasts.

Human gingival fibroblasts and oral epithelial cells serve as the first line of defense against periodontal infection [42]. Their roles in maintaining oral health and responding to microbial and masticatory challenges are crucial for preventing and managing periodontal diseases [43,44]. They were identified to be significant drivers of inflammation in periodontitis [8] and periimplantitis [9] because of their active participation in modulating the local inflammatory environment. The cells releasing chemokines recruit and activate immune cells, such as neutrophils, macrophages, and lymphocytes, at the inflammatory sites. Their respective roles and relevance drive the choice of gingival fibroblasts and HSC2 cells in periodontal research to different aspects of the disease. Gingival fibroblasts offer insights into connective tissue responses to inflammation and healing [8], while HSC2 cells help understand epithelial cell behavior and pathology [8,44]. Considering our previous study showing that the cell source and species matter in a bioassay, human gingival fibroblasts and HSC2 cells were utilized. Together, we provide a robust model for investigating the complex interactions and mechanisms underlying periodontal diseases.

Clinically, DMF is commonly used in the treatment of chronic inflammatory diseases, such as relapsing multiple sclerosis and psoriasis, under the trade name Tecfidera^®^ [11] and Skilarence^®^ [12], but is not limited to these pathologies [13]. The clinical advantages of DMF compared to other drugs such as NSAIDs and corticosteroids might include the safety profile; considering its clinical use, DMF has a low risk of serious adverse effects. Moreover, our hydrogel approach is designed as an injectable DMF-releasing concept targeting local cells in an inflammatory environment. Consistently, in vitro, DMF can reduce a provoked inflammatory response in fibroblasts [14], smooth muscle cells [15], endothelial cells [16,17], epithelial cells [18], chondrocytes [19], pulposus cells [45], and microglia [46]. In the context of dentistry, DMF lowered pyroptosis in gingival and pulp fibroblasts [22]. In oral squamous cell carcinoma cell lines, DMF modulates apoptosis, oxidative stress, and epithelial-mesenchymal transformation [23]. In the present study, DMF effectively reduced the expression of CXCL8, CXCL1, CXCL2, and CCL2 without affecting cell viability in primary gingival fibroblasts or HSC2 cells. There was also a trend where DMF lowered the expression of other cytokines and chemokines, namely IL6, CCL4, CCL2, and CCL20. Thus, our observation supports the existing knowledge that, at least at rather high concentrations, DMF lowers chemokine expression in oral cells. This robust activity has prompted us to test the activity of DMF released from gellan gum hydrogel. Indeed, DMF-loaded hydrogels were capable of reducing the forced expression of CXCL1, CXCL2, CXCL8, and CCL2 in gingival fibroblasts. These findings suggest that DMF released from the hydrogel scaffold remains active.

Gellan gum hydrogels provide a three-dimensional polymer network capable of absorbing, retaining, and releasing bioactive molecules; for instance, caffeic acid phenethyl ester [30], dexamethasone [31], piroxicam [32], naproxen [33], and diclofenac [34]. The release profile can be finely tuned by manipulating the crosslinking density, polymer composition, and environmental triggers such as pH, temperature, or enzymes [47,48]. Gellan gum hydrogels are therefore beneficial for localized delivery of drugs such as DMF with narrow therapeutic windows—exemplified by dental applications. Gellan gum hydrogels carrying DMF might be injected into inflamed periodontal pockets where they exert their anti-inflammatory activity; the same is true for periimplantitis. With this clinical scenario in mind, we tested DMF-loaded hydrogels in cocultures with oral cells in a simulated inflammatory environment—showing that DMF-loaded hydrogels could reduce the induced expression of CXCL1, CXCL2, CXCL8, and CCL2 in gingival fibroblasts. These findings provide a solid support for future research to test for in vivo efficiency of DMF-loaded gellan gum hydrogels in periodontitis or periimplantitis. The question arises over the differences and similarities of the DMF-gellan gum system presented in this study with other DMF drug delivery systems, such as nanoparticles or nanogels; for instance, platelet nanogels [49], chitosan-alginate core-shell-corona-shaped nanoparticles [50], lipid [51], and transethosomes [52]. They have received significant attention as drug delivery systems due to their unique properties, such as high drug loading capacity, biocompatibility, and ability to deliver drugs in a controlled manner. However, there are limitations, such as complex chemical processes, difficulty in scaling up, issues with physical–chemical stability, unpredictable biodistribution, and lengthy and complicated regulatory approval. Hydrogels, however, offer numerous advantages as drug delivery systems, including biocompatibility, controlled and sustained release, versatility in drug encapsulation, and the ability to provide localized and minimally invasive delivery. These properties make hydrogels, like the FDA-approved polysaccharide polymer gellan gum, a powerful platform for enhancing the efficacy and safety of various therapeutic agents, potentially improving patient outcomes and quality of life.

Our in vitro study has limitations; for example, even though our findings support the principal activity of DMF in dampening forced-chemokine expression by gingival fibroblast and HSC2 cells, the underlying molecular mechanisms remain to be discovered. The research remains on a descriptive level considering signaling, as we have limited the study to MAPK and NFκB-p65 phosphorylation. Future research should consider a phosphorylation screening assay to identify other signaling pathways modulated by DMF, as it is used for drug screening [53]. Moreover, future studies might involve RNAseq and other omics approaches to learn more about DMF target genes and signaling mechanisms that are modulated by the drug. Another limitation is that the gellan gum hydrogel alone provokes a cellular response that is not understood. Considering this premise, we have developed a bioassay where the hydrogel was allowed to solidify in the border of the tissue culture well, only covering a small area of the cell layer. Cells not covered by the gellan gum hydrogel could respond to DMF released into the culture medium. Our data support the assumption that DMF released from the hydrogel maintains its activity by lowering chemokine expression in vitro. Future in vivo research should aim to develop more effective strategies for implementing gellan gum hydrogels loaded with DMF for managing oral inflammatory diseases, including periodontitis and peri-implantitis.

## 4. Materials and Methods

### 4.1. Gingival Fibroblasts and Oral Squamous Cell Carcinoma Cells HSC2

Human gingival fibroblasts were isolated from explant cultures of three independent donors after approval of the Ethics Committee of the Medical University of Vienna (EC No. 631/2007). Gingival fibroblasts were seeded at a density of 30,000/cm^2^ onto culture plates one day before stimulation. The oral squamous cell carcinoma cell line HSC2 was obtained from the Health Science Research Resources Bank (Sennan, Japan) and seeded at 60,000/cm^2^ one day before stimulation. All cells were grown in Dulbecco’s modified Eagle’s medium (DMEM, Sigma-Aldrich, St. Louis, MO, USA) supplemented with 10% fetal calf serum (Bio & Shell GmbH, Nuremberg, Germany) and antibiotics (Sigma, St. Louis, MO, USA).

### 4.2. Dimethyl Fumarate and Inflammatory Agonist Stimulations

In the primary setting, gingival fibroblasts and HSC2 cells were treated overnight with 10 ng/mL IL1β and 10 ng/mL TNFα (both ProSpec, Ness-Ziona, Israel), or 30% sterile saliva (SLV) [36], respectively, with and without dimethyl fumarate (DMF, Sigma Aldrich, St. Louis, MO, USA) or with serum-free medium alone at 37 °C, 5% CO_2_, and 95% humidity before analysis. Whole human saliva was collected from the authors (L.W. and N.d.S.S.), who are non-smokers and gave their informed consent. Salival flow was stimulated by chewing paraffin wax (Ivoclar Vivadent AG, Schaan, Liechtenstein) without eating or drinking for 1 h before collection. Immediately after collection, the saliva was centrifuged at 4000× *g* for 5 min. The saliva supernatant was passed through a filter with a pore diameter of 0.2 µm (Diafil PS, DIA-Nielsen GmbH, Düren, Germany). Frozen stocks were prepared. Saliva was used solely to provoke an inflammatory response.

### 4.3. Dimethyl Fumarate-Loaded Gellan Gum Hydrogel

Dimethyl fumarate (DMF) was dissolved in dimethyl sulfoxide at the stock concentration of 200 mM. Gellan gum powders were dissolved in distilled water by warming up to 90 °C for 30 min until becoming transparent. The DMF-loaded hydrogel was prepared by mixing the gellan gum precursors and the DMF stock solution in following formulations: LG: 1 mL gellan gum + 2.5 μL DMF (the DMF in hydrogel was 0.5 mM); HG: 1 mL gellan gum + 5 μL DMF (the DMF in hydrogel was 1 mM); while GG was gellan gum alone. As a bioassay, 120 μL of DMF-loaded hydrogel precursors were placed perpendicular to the wall of a 24-well plate seeded with gingival fibroblasts, followed by adding 600 μL of serum-free media.

### 4.4. Physicochemical Properties of Gellan Gum Hydrogel

The rheological study of the hydrogels was performed using a rheometer (DHR-2, TA Instruments, New Castle, DE, USA) following a previous study [54]. The storage modulus (G′) and loss modulus (G″) of the hydrogels were tested in oscillatory mode as a function of temperature at a constant strain of 1% and a fixed oscillation frequency of 1 Hz using a parallel plate configuration (8 mm in diameter). The morphology of the hydrogels was observed using a scanning electronic microscope (SEM, SU8010, Hitachi, Tokyo, Japan). Lyophilized hydrogels were sputter-coated with gold for 60 s before the observation. The swelling and degradation behavior were determined by immersing the hydrogels in PBS at 37 °C (physical temperature of the mouth). The sample mass was weighed at pre-determined time points over 5 days. The mass change of the hydrogel was calculated using the following equation: mass change (%) = (m_t_ − m_0_)/m_0_ × 100%, where m_t_ is the weight of the hydrogel at time t and m_0_ is the initial weight of the hydrogel. The mass change was presented as the mean ± standard deviation of three separate measurements.

### 4.5. Release Kinetics of Dimethyl Fumarate from Gellan Gum Hydrogel

The in vitro release of DMF from hydrogels loaded with 5.0 mM (HG) and 2.5 mM (LG) was studied by using a dialysis bag with a molecular weight cut-off of 3500 Da. The dialysis bags were placed in 20 mL of phosphate-buffered saline (PBS, pH = 7.4) in an air shaker at 37 °C while stirring at 100 rpm. At the indicated time points, 2 mL of release medium was collected and replaced with fresh PBS. The release medium was subjected to high-performance liquid chromatography (HPLC; Waters Alliance 2695; Milford, MA, USA) on a reverse-phase C18 column (4.6 mm × 150 mm, 5 mm, Sunfire Analysis column; Milford, MA, USA) at room temperature. The mobile phase was acetonitrile, degassed and filtered. The flow rate was 0.8 mL/min and the eluent was detected using a Waters PDA detector at 210 nm. The cumulative release (%) = m_t_/m_0_ × 100%, where m_t_ is the total amount of the DMF at time point t and m_0_ is the initial amount of the loaded DMF.

### 4.6. Cell Viability Assay

The cells were exposed to 12.5–200 μM of DMF in serum-free media. After 24 h, 0.5 mg/mL MTT (Sigma Aldrich, St. Louis, MO, USA) was added to each well and placed in the incubator for 2 h. After removing the medium, formazan crystals were solubilized with dimethylsulphoxide (Sigma Aldrich). The optical density was measured at 570 nm. The data from independent experiments are presented as percentages of the optical density in the treatment groups, normalized to the unstimulated control.

### 4.7. Trypan Blue Staining and Live/Dead Staining

After stimulation of the cells for 24 h, trypan blue was applied to test the cellular membrane integrity. Trypan blue (0.4%, Sigma Aldrich, St. Louis, MO, USA), diluted 1:1 in a phosphate buffer (PBS, pH 7.4), was added to each well and incubated for 2–3 min at room temperature. The trypan blue was discarded and the cells were examined using light microscopy. A live/dead staining assay kit was also employed to further confirm cell viability. The protocols were followed according to the manufacturer’s instructions (Enzo Life Science, Inc., Lausanne, Switzerland).

### 4.8. Real-Time Polymerase Chain Reaction (RT-PCR) and Immunoassay

The total RNA was isolated with the ExtractMe total RNA kit (Blirt S.A., Gdańsk, Poland) according to the manufacturer’s protocol, followed by reverse transcription and a polymerase chain reaction (LabQ, Labconsulting, Vienna, Austria) on a CFX Connect^TM^ Real-Time PCR Detection System (Bio-Rad Laboratories, Hercules, CA, USA). The mRNA levels were calculated by normalizing to the housekeeping gene GAPDH using the ∆∆Ct method. The primer sequences are shown in Table 1. For the immunoassay, the human CXCL8 kit (DY208-R&D Systems, Minneapolis, MN, USA) was used according to the manufacturer’s introductions.

### 4.9. Immunofluorescence Analysis

Gingival fibroblasts were cultured onto the Millicell^®^ EZ slides (Merck KGaA, Darmstadt, Germany) with a density of 10,000 cells/cm^2^ and the serum was starved overnight. The cells were stimulated with 25–100 µM DMF or 10 ng/mL IL1β and 10 ng/mL TNFα for 1 h using a serum-free medium as a control. The cells were fixed with 4% paraformaldehyde, blocked with 1% bovine serum albumin (BSA), and permeabilized with 0.3% Triton X-100 (all Sigma-Aldrich). The anti-NF-κB P65 antibody (Cell Signaling Technology, CST, Cambridge, UK, #8242) was used overnight at 4 °C. Detection was performed with an Alexa 488 secondary antibody (CS-4412, CST). Images were taken by a fluorescence microscope with the DAPI-FITC dual excitation filter block (Echo Revolve Fluorescence Microscope, San Diego, CA, USA).

### 4.10. Western Blot

Gingival fibroblasts were serum starved overnight and exposed to 100 μM DMF or 10 ng/mL IL1β and 10 ng/mL TNFα for 1 h. Cell extracts containing SDS buffer with protease and phosphatase inhibitors (Complete Ultra Tablets and PhosStop; Roche, Mannheim, Germany) were separated by SDS–PAGE and transferred onto PVDF membranes (Roche Diagnostics, Mannheim, Germany). The membranes were blocked and the binding of the primary antibodies phosphor-NFκB-p65 and NFκB-p65 (Cell Signaling Technology; #3033, #8242), phosphor-p38 and p38 (Santa Cruz Biotechnology, SCBT; #4511, #535), phosphor-ERK and ERK (SCBT; #7383, #81459), and phosphor-JNK and JNK (SCBT; #6254, #7345) were detected with the appropriate secondary antibody labeled with HRP. Membranes were incubated with Clarity Western ECL Substrate (Bio-Rad Laboratories, Inc., Hercules, CA, USA) for 5 min before chemiluminescence signals were visualized with a ChemiDoc imaging system (Bio-Rad Laboratories).

### 4.11. Statistical Analysis

All experiments were performed at least four times. Statistical analyses were performed with ratio-paired *t*-tests. Analyses were performed using Prism v.9 (GraphPad Software; San Diego, CA, USA). Significance was set at *p* < 0.05.

## 5. Conclusions

In closing this report, we provide in vitro evidence that DMF lowers chemokine’s forced expression in oral cell bioassays, and also when released from self-hardening gellan gum hydrogels. This research is a step towards the translation of our findings into a clinical scenario in dentistry for local inflammation, where gellan gum hydrogels carrying DMF can be applied.

## Figures and Tables

**Figure 1 ijms-25-09485-f001:**
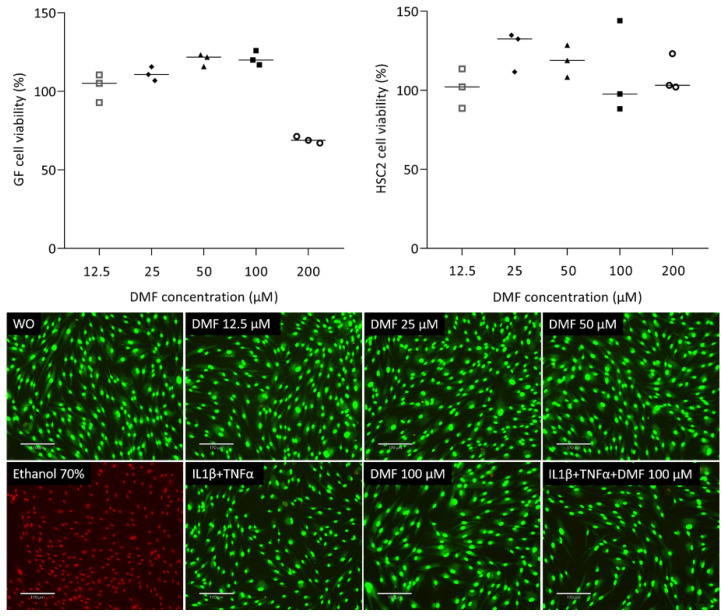
Cell viability of gingival fibroblasts and HSC2. Cells were exposed overnight to the serum-free media with a serial dilution of DMF and formazan formation is normalized to untreated control cells. Representative live/dead staining of gingival fibroblasts allows to distinguish green living from red dead cells. Cells were grown with and without DMF in the presence of IL1β and TNFα overnight and subjected to live/dead staining. The scale bar represents 170 μm.

**Figure 2 ijms-25-09485-f002:**
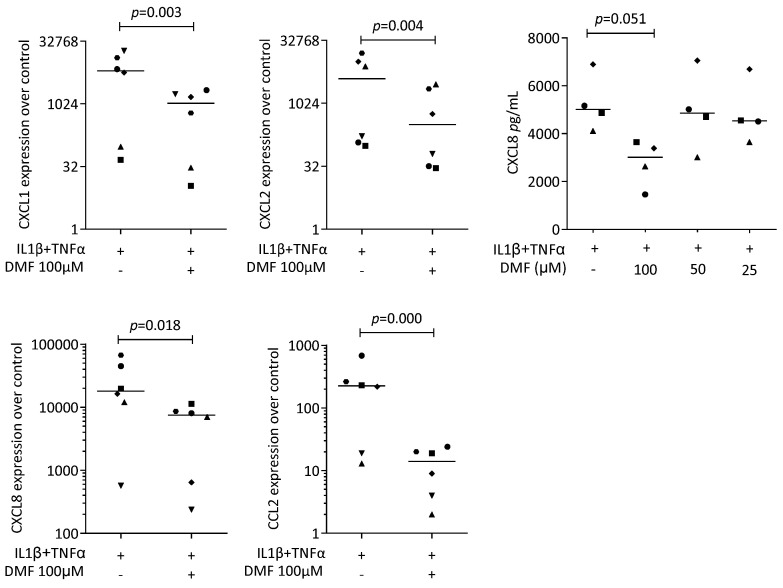
Gene expression of chemokines in gingival fibroblasts exposed to IL1β and TNFα. A total of 100 μM DMF significantly reduced chemokine expression. Data are expressed as x-fold over the respective untreated controls. Data points represent six and four independent experiments for PCR and ELISA, respectively. Statistical analysis was based on ratio-paired *t*-tests, and *p*-values are indicated.

**Figure 3 ijms-25-09485-f003:**
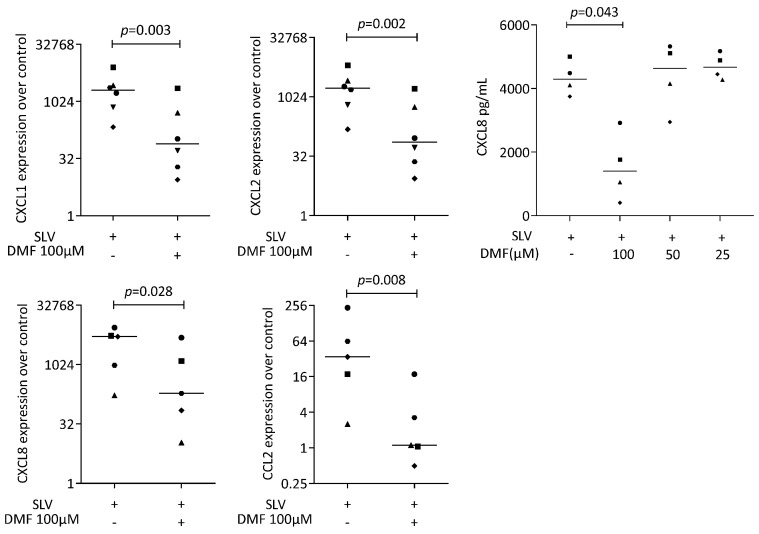
Gene expression of chemokine in gingival fibroblasts under saliva (SLV). A total of 100 μM DMF significantly reduced chemokine expression. Data are expressed as x-fold over the respective untreated controls. Data points represent six and four independent experiments for PCR and ELISA, respectively. Statistical analysis was based on ratio-paired *t*-tests, and *p*-values are indicated.

**Figure 4 ijms-25-09485-f004:**
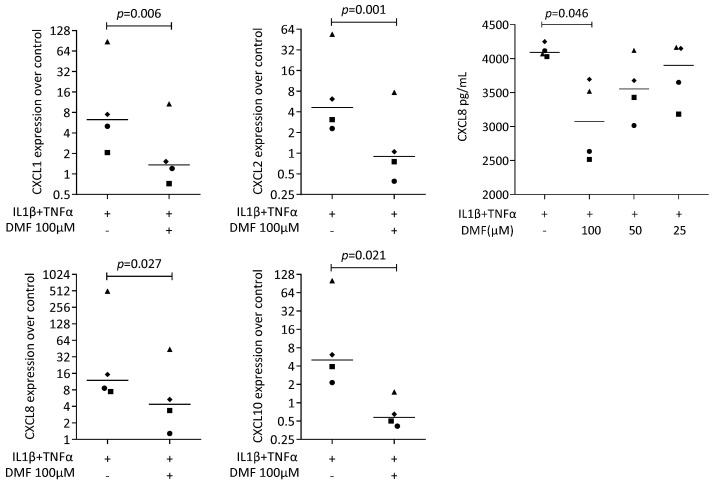
Gene expression of chemokine in HSC2 cells under IL1β and TNFα. A total of 100 μM DMF significantly reduced chemokine expression. Data are expressed as x-fold over the respective untreated controls. Data points represent six and four independent experiments for PCR and ELISA, respectively. Statistical analysis was based on ratio-paired *t*-tests, and *p*-values are indicated.

**Figure 5 ijms-25-09485-f005:**
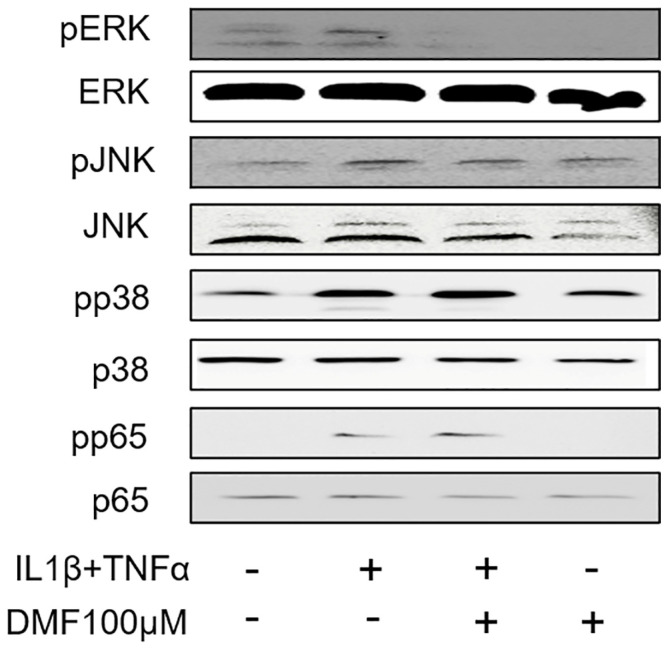
Effects of DMF on phosphorylation of signaling molecules in IL1β and TNFα-stimulated gingival fibroblasts. The cell lysates were used for the detection of phosphorylated or total extracellular signal-regulated kinase (ERK), c-Jun N-terminal kinase (JNK), p38, and p65 via Western blotting.

**Figure 6 ijms-25-09485-f006:**
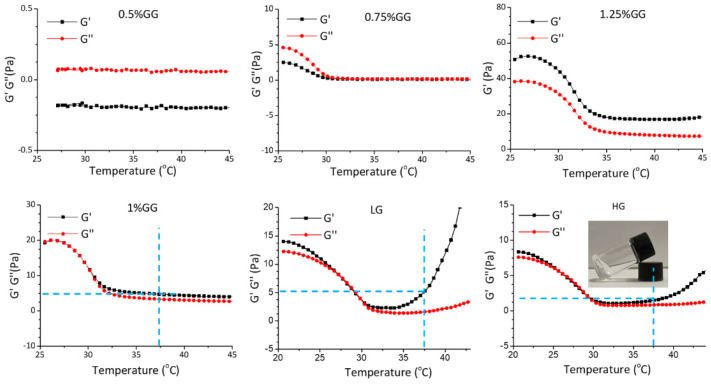
Rheological properties of the hydrogels. Temperature-sweep measurements of the hydrogels with different components. 0.5%GG: 5 mg/mL gellan gum hydrogel, 0.75%GG: 7.5 mg/mL gellan gam hydrogel, 1.25%GG: 12.5 mg/mL gellan gum hydrogel, 1%GG: 10 mg/mL gellan gum hydrogel, LG: 10 mg/mL gellan gum hydrogel with 0.5 mM DMF, HG: 10 mg/mL gellan gum hydrogel with 1 mM DMF.

**Figure 7 ijms-25-09485-f007:**
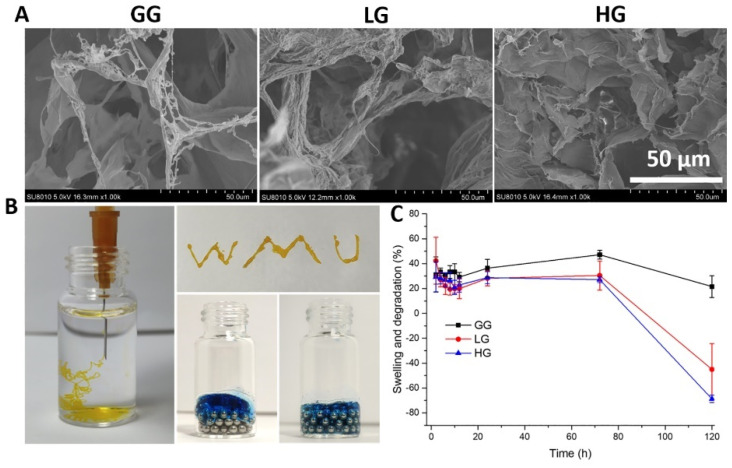
(**A**) Representative SEM images of the gellan gum hydrogels with and without DMF; (**B**) injectability of the hydrogels; (**C**) swelling and degradation properties of the hydrogel. GG: gellan gum hydrogel alone; LG: gellan gum hydrogel with 0.5 mM DMF, HG: gellan gum hydrogel with 1 mM DMF.

**Figure 8 ijms-25-09485-f008:**
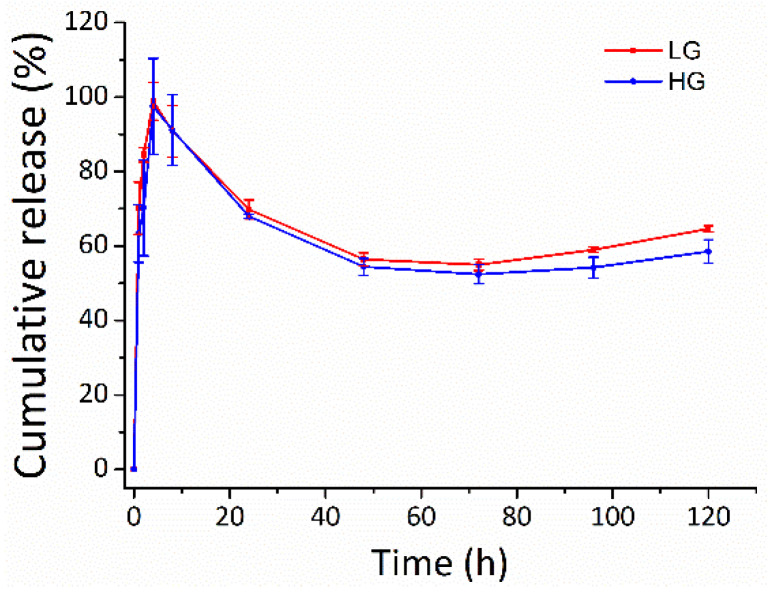
The release profile of the DMF-loaded hydrogels. Long-term slow-release kinetics of DMF from the gellan gum hydrogel in both LG and HG groups. LG: gellan gum hydrogel with 2.5 mM DMF, HG: gellan gum hydrogel with 5 mM DMF.

**Figure 9 ijms-25-09485-f009:**
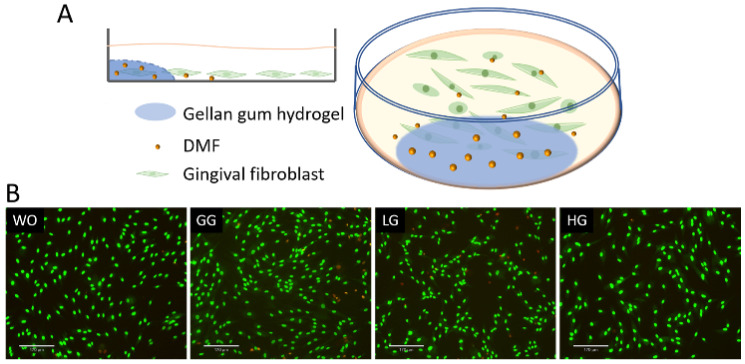
(**A**) Schematic diagram of gingival fibroblast cocultured with hydrogel; (**B**) representative images of live/dead staining of gingival fibroblasts with hydrogels. Cells were grown with DMF-loaded hydrogels overnight and subjected to live/dead staining. WO: without treatment; GG: gellan gum hydrogel alone; LG: gellan gum hydrogel with 0.5 mM DMF, HG: gellan gum hydrogel with 1 mM DMF. Scale bars represent 170 μm.

**Figure 10 ijms-25-09485-f010:**
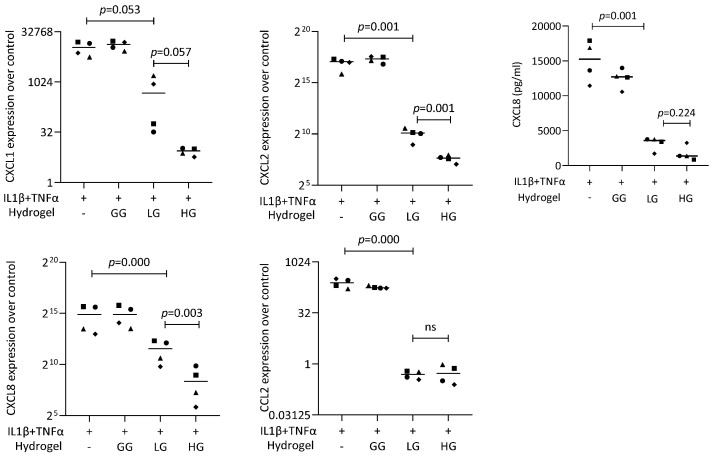
Gene expression of chemokines in gingival fibroblasts under IL1β and TNFα. HG and LG significantly reduced chemokine expression. Data are expressed as x-fold over the respective untreated controls. Data points represent four independent experiments for PCR and ELISA. Statistical analysis was based on ratio-paired *t*-tests and *p*-values are indicated.

**Table 1 ijms-25-09485-t001:** The primer sequences.

Genes	Forward Sequences	Afterward Sequences
*CXCL1*	TCCTGCATCCCCCATAGTTA	CTTCAGGAACAGCCACCAGT
*CXCL2*	CCCATGGTTAAGAAAATCATCG	CTTCAGGAACAGCCACCAAT
*CXCL8*	AACTTCTCCACAACCCTCTG	TTGGCAGCCTTCCTGATTTC
*CXCL10*	TGCCATTCTGATTTGCTGCC	TGCAGGTACAGCGTACAGTT
*CCL2*	AGAATCACCAGCAGCAAGTGTC	TCCTGAACCCACTTCTGCTTG
*GAPDH*	AAGCCACATCGCTCAGACAC	GCCCAATACGACCAAATCC

## Data Availability

The original contributions presented in the study are included in the article/Appendix A. Further inquiries can be directed to the corresponding author.

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
