# Peer review of "Dimethyl Fumarate-Loaded Gellan Gum Hydrogels Can Reduce In Vitro Chemokine Expression in Oral Cells"

_ijms, 2024, doi:10.3390/ijms25179485_

Round 1

Reviewer 1 Report

Comments and Suggestions for Authors

This paper is about the control of inflammation in periodontitis by DMF. This attempt is new and original. However, there are several issues that need to be resolved.

The figures are small and difficult to understand.

The growth study is being conducted, but cytotoxicity should also be considered.

As mentioned in the discussion section, the mechanism of action of DMF to suppress chemokines should also be examined. For example, the MSPK and NFkB pathways. The content of this paper would be considered somewhat low quality.

Also, in the discussion section, it is necessary to add the clinical advantages of DMF compared to other drugs.

Author Response

Authors’ Response to the comments of Reviewer #1

  1. This paper is about the control of inflammation in periodontitis by DMF. This attempt is new and original. However, there are several issues that need to be resolved.

  1. The figures are small and difficult to understand.

Answer: We thank for the feedback. We made the necessary adjustments to ensure the figures were larger and more comprehensible.

  1. The growth study is being conducted, but cytotoxicity should also be considered.

Answer: We did the cytotoxicity of DMF and gellan gum hydrogel with DMF as shown in Figure 1 and Figure 8, respectively. Our results showed that DMF 100 μM and lower was nontoxic to oral cells, and the DMF-loaded gellan gum hydrogel was safe for the oral cells. We were surprised by the sharp change of formazan formation between 100 and 200 µM DMF in the fibroblasts – but our data agree with other studies mostly recommending to work with 100 µM DMF in vitro. We have slightly adapted the text.

  1. As mentioned in the discussion section, the mechanism of action of DMF to suppress chemokines should also be examined. For example, the MAPK and NFkB pathways. The content of this paper would be considered somewhat low quality.

Answer: Yes, we fully agree that our research remains on the descriptive level considering signaling; we show in the Supplement the DMF failed to lower IL1/TNF-induced p65 phosphorylation and nuclear translocation changes, as well as the phosphorylation of p38. We could have done a phosphorylation screening array to identify which signaling pathway is inhibited by DMF; but in the revised version we consider this as study limitation and we recommend doing this analysis in future studies.

  1. Also, in the discussion section, it is necessary to add the clinical advantages of DMF compared to other drugs.

Answer: Thank you very much for this suggestion; it is not easy to compare drugs but we have discussed the relevance considering the good safety profile compared to NSAIDs and corticosteroids. This aspect is now written in the discussion.

Reviewer 2 Report

Comments and Suggestions for Authors

The manuscript entitled "Dimethyl fumarate loaded gellan gum hydrogels can reduce in vitro chemokine expression in oral cells" describes in vitro studies regarding the use of dimethyl fumarate loaded gellan gum hydrogels as drug delivery systems in the treatment of oral inflammatory diseases, including periodontitis and peri-implantitis. The discovery of efficient drug delivery systems used in the treatment of periodontal disease is welcome because according to the World Health Organization this disease affects three quarters if the world's population.

The title of the manuscript is well-aligned with the content.

The experimental part is comprehensive and well-structured.

However, the manuscript could benefit from a more detailed discussion, especially a comparative analysis with other studies existing in literature is needed, highlighting the difference and similarities of the DMF-gellan gum system presented in this study with other similar DMF drug delivery systems.

Also, some graphics are of low quality (Figures 2,3,4 and 9) being difficult to follow.

At the end, a Conclusions Section is necessary.

Author Response

Authors’ Response to the comments of Reviewer #2

  1. Dimethyl fumarate (DMF), originally proposed to treat multiple sclerosis, was considered to have a spectrum of anti-inflammatory effects that effectively control periodontitis, mainly when applied with a hydrogel delivery system. The authors provided in vitro evidence that DMF lowers chemokine's forced expression in oral cell bioassays, also when released from self-hardening gellan gum hydrogels. This contribution is well organized. Nevertheless, the following issue should be resolved before further consideration.

Answer: We thank the reviewer for the positive comments. The issues were answered as follows.

  1. Figure 7. Why was the "cumulative" release at 24 hours less than the value at 10 or 12 hours? I guess the authors should properly check the definition of "cumulative" release. Further, why did the GG show the "cumulative" release of DMF? Should the GG contain no DMF at all?

Answer: We thank the reviewer for the careful reading. Cumulative release was defined as follows: Cumulative release (%) = mt/m0 X 100%, where mt was the total mass amount of the DMF at time point t, m0 was the initial amount of the loaded DMF. As suggested by the reviewer, we checked the cumulative and redid the figure, as shown Figure 7. We deleted the GG group; it was a misunderstanding. The reason why the cumulative release at 24 hours less than that at 10 or 12 hours could be the rapid hydrolysis of DMF to monomethyl fumarate (MMF) in phosphate buffer, pH 7.4 according to previous studies. We have added this in the text.

Page 9:  see Figure in the text

Figure 7. The release profile of the DMF-loaded hydrogels. Long-term slow release kinetic of DMF from the gellan gum hydrogel in both LG and HG groups. LG: gellan gum hydrogel with 2.5 mM DMF, HG: gellan gum hydrogel with 5 mM DMF.

Reviewer 3 Report

Comments and Suggestions for Authors

Dimethyl fumarate (DMF), originally proposed to treat multiple sclerosis, was considered to have a spectrum of anti-inflammatory effects that effectively control periodontitis, mainly when applied with a hydrogel delivery system. The authors provided in vitro evidence that DMF lowers chemokine's forced expression in oral cell bioassays, also when released from self-hardening gellan gum hydrogels.

This contribution is well organized. Nevertheless, the following issue should be resolved before further consideration

1.      Figure 7. Why was the "cumulative" release at 24 hours less than the value at 10 or 12 hours? I guess the authors should properly check the definition of "cumulative" release. Further, why did the GG show the "cumulative" release of DMF? Should the GG contain no DMF at all?

Author Response

Authors’ Response to the comments of Reviewer #3

  1. The manuscript entitled "Dimethyl fumarate loaded gellan gum hydrogels can reduce in vitro chemokine expression in oral cells" describes in vitro studies regarding the use of dimethyl fumarate loaded gellan gum hydrogels as drug delivery systems in the treatment of oral inflammatory diseases, including periodontitis and peri-implantitis. The discovery of efficient drug delivery systems used in the treatment of periodontal disease is welcome because according to the World Health Organization this disease affects three quarters if the world's population.
  2. The title of the manuscript is well-aligned with the content.
  3. The experimental part is comprehensive and well-structured.

Answer: We thank the reviewer for the above positive comments.

  1. However, the manuscript could benefit from a more detailed discussion, especially a comparative analysis with other studies existing in literature is needed, highlighting the difference and similarities of the DMF-gellan gum system presented in this study with other similar DMF drug delivery systems.

Answer: We thank the reviewer for the suggestion. We added some sentences of the detailed discussion about the current DMF delivery system and highlighted the difference and similarities.

  1. Also, some graphics are of low quality (Figures 2,3,4 and 9) being difficult to follow.

Answer: We thank for the feedback. We made the necessary adjustments to ensure the figures were larger and more comprehensible.

  1. At the end, a Conclusions Section is necessary.

Answer: We thank the reviewer for the feedback. The conclusion section was the last paragraph of the Discussion. We added a Conclusion before this paragraph.

Round 2

Reviewer 1 Report

Comments and Suggestions for Authors

The authors have overcome some of the requirements, but some remain unresolved.

Investigation of the mechanisms of action of MAPKs, NFkB, etc. by Western blotting is an essential process for a better understanding of molecular biology.

I would highly recommend that this issue be resolved.

Author Response

Reviewer 1:

The authors have overcome some of the requirements, but some remain unresolved. Investigation of the mechanisms of action of MAPKs, NFkB, etc. by Western blotting is an essential process for a better understanding of molecular biology.I would highly recommend that this issue be resolved.

Answer: We appreciate the reviewer’s suggestion to include an investigation of the molecular mechanisms. At the very beginning, we did western blotting for a better understanding of molecular mechanism, using PP65, P65, PP38, and P38 as markers for NFkB and MAPK pathway, respectively. The results were shown in supplementary materials Figure 3. In response to the reviewer, we have conducted additional experiments to explore these mechanisms further. We used PERK, ERK, PJNK, JNK, and β-Actin as markers, respectively. The results showed DMF works with ERK-MAPK pathway. These new findings have been incorporated into the main manuscript, with the relevant sections highlighted in red.

Reviewer 2:

The manuscript has been modified according to the requirements of the reviewers.

Reviewer 3:

The authors have properly addressed the issues raised from earlier version.

Answer: We thank the reviewers for their recognition.

Reviewer 2 Report

Comments and Suggestions for Authors

The manuscript has been modified according to the requirements of the reviewers.

Author Response

Reviewer 2:

The manuscript has been modified according to the requirements of the reviewers.

Answer: We thank reviewer for the work.

Reviewer 3 Report

Comments and Suggestions for Authors

The authors have properly addressed the issues raised from earlier version.

Author Response

Reviewer 3:

The authors have properly addressed the issues raised from earlier version.

Answer: We thank the reviewers for their recognition.